# Conductive MoO₃–PEDOT:PSS Composite Layer in MoO₃/Au/MoO₃–PEDOT:PSS Multilayer Electrode in ITO-Free Organic Solar Cells

Md Maniruzzaman [1,2], Rahim Abdur [2,3], Md Abdul Kuddus Sheikh [2,4], Son Singh [2] and Jaegab Lee [2,*]

1 Department of Chemistry, Khulna University of Engineering and Technology, Khulna 9203, Bangladesh
2 School of Advanced Materials Engineering, Kookmin University, Seoul 136-702, Republic of Korea
3 Institute of Fuel Research and Development, Bangladesh Council of Scientific and Industrial Research, Dhaka 1205, Bangladesh
4 Łukasiewicz Research Network-PORT Polish Center for Technology Development, 54-066 Wrocław, Poland
* Correspondence: lgab@kookmin.ac.kr

**Abstract:** The solution-processed and conductive MoO₃–PEDOT:PSS (Mo–PPSS) composite layer in a MoO₃/Au/MoO₃–PEDOT:PSS (MoAu/Mo–PPSS) multilayer electrode in ITO-free organic solar cells (OSCs) was optimized in terms of electrical conductivity, interfacial contact quality, work function, and process wettability of the conductive composite thin film. The surface composition of the PEDOT:PSS film onto different electrodes was observed by using X-Ray Photoelectron Spectroscopy. The PEDOT:PSS-MoO₃ composite protects the dissolution of individual MoO₃ with PEDOT:PSS, which was confirmed by Auger Electron Spectroscopy. The UV-Visible spectroscopy showed that the photoactive layer of P3HT:PCBM absorbs in the wavelength range of 300–650 nm with the maximum absorption at 515 nm (2.40 eV). The device performance of 3.97% based on an MoAu/Mo–PPSS conductive composite electrode exhibited comparable enhancement and only 6% enhancement compared to an ITO-based electrode (3.91%). The enhancement of device efficiency was mainly due to relatively higher conductivity, a low work function of the conductive metal oxide-metal-metal oxide/polymer composite, and an enhancement of interfacial contact quality between the hole transport layer (HTL) and the mixed organic polymeric photoactive layer. These results indicate that the solution-processable Mo–PPSS conductive composite layer of the MoO₃/Au multilayer electrode can replace the ITO-based electrode in the bulk of heterojunction organic photovoltaics (OPVs).

**Keywords:** conductive composite; multilayer electrode; ITO-free; organic solar cells; MoO₃

## 1. Introduction

Conductive and transparent electrodes are imperative for efficient organic optoelectronic devices, such as organic solar cells (OSCs), organic thin film transistors (OTFTs), and organic light-emitting diodes (OLEDs) [1,2]. Due to its higher transparency in the visible wavelength range and good electrical conductivity, conventional indium tin oxide (ITO) has been commonly used as a transparent conducting electrode in optoelectronic applications [3]. However, ITO has some intrinsic drawbacks as a conductive and transparent electrode, due to its poor mechanical stability [4], the scarcity of indium [5–8], and the highly expensive cost of high-temperature vacuum deposition methods for flexible electrodes [9]. For these reasons, promising ITO alternatives, such as carbon nanotubes [10], graphene [11], roll-to-roll [12], conductive polymer transparent electrodes [13–16], semi-transparent electrodes [17], and metallic nanowires [18], have been proposed as a transparent electrode.

Recently, one of the most common approaches applied to achieve the advantages of high optical transmittances of the metal oxides (MOs), band offsets in the energy diagram, and the magnificent conductivity of metals in the form of a thin metal layer between two metal oxide layers (MO/M/MO) has been used for the replacement of ITO [19–24]. Due to

their high transparency, high hole affinity, high work function (WF), and easy as well as simple solution process ability, the poly(3,4-ethylenedioxythiophene): poly(styrenesulfonate) (PEDOT:PSS) thin films have been commonly used as the hole transport layer (HTL) [25,26]. However, the poor stability of conductive metal oxides (TCO) is due to contact with the highly acidic and hygroscopic nature of PEDOT:PSS film, which might limit its applications to various devices by reducing the lifetime of the devices [27,28]. It has long been assumed that the highly acidic nature of the PEDOT:PSS layer will etch away the underlying metal oxides, therefore significantly reducing the prepared device performance, and that the hygroscopicity of PEDOT:PSS is also detrimental to prepared device stability [29,30]. To overcome these issues, some metal oxide thin layers, e.g., Molybdenum trioxide ($MoO_3$) [31], Zinc Oxide (ZnO) [32], and Vanadium pentoxide ($V_2O_5$) [33], have been used to replace highly acidic and hygroscopic PEDOT:PSS in polymer solar cells (PSCs). An Aluminum Oxide ($Al_2O_3$) interlayer was used as a protective layer to protect the dissolution of thin layer $MoO_3$ with the PEDOT:PSS layer, but the device performance was only 2.77% in our previous work [28]. For low-cost and large-area deposition, the solution-processed $MoO_3$ is a promising candidate to be used as HTL in PSCs, as well as other types of photovoltaic devices. The solution-processed $MoO_3$ can be prepared from acidic precursors [34], as well as from $MoO_3$ and PEDOT:PSS composites [35,36]. Moreover, the poor conductivity of $MoO_3$ is the main question for low-cost and large-area applications [37]. Therefore, it is essential to develop easy solution-processable, highly conductive composite materials (metal oxide/polymer) for efficient PSCs. Moreover, these metal oxide/polymer composites can improve interfacial interaction and facilitate faster and more effective charge transfers from the polymeric photoactive layers to the electrodes.

In the case of OSCs, the most popular and common active layer is either the bilayer or heterogeneous mixture of poly(3-hexylthiophene) (P3HT) and fullerene derivative [6,6] phenyl-C61-butyric acid methyl ester (PCBM), or non-fullerene PCBM. Among them, heterogeneous mixtures are more favorable due to their rapid charge transfer and fewer recombinations of excitons [13,38,39]. As a photoactive layer, fullerene or non-fullerene PCBM [38,39] are commonly used with regioregular poly(3-hexylthiophene) (P3HT), where P3HT acts as a photoactive electron donor due to its structural, photophysical, laminar crystal structure that enables high hole mobility (>$10^{-2}$ cm$^2$/Vs) [40], despite the mismatch of its absorption in the solar spectrum. To minimize the mismatch of P3HT absorption, PCBM is commonly used due to its ball-like fully conjugated structures, which have strong electron affinity and unipolar electron transport that promotes the delocalization of electrons [41].

In this study, a solution-processable and unprejudiced $MoO_3$–PEDOT:PSS conductive composite (Mo–PPSS conductive) was utilized on the $MoO_3$/Au (MoAu) structure to fabricate a MoAu/Mo–PPSS conductive multilayer ITO-free electrode on the glass substrate, and the conductivity, transmittance, work function, and elemental chemical composition were investigated. The surface composition of the PEDOT:PSS film onto different electrodes was studied by using X-ray Photoelectron Spectroscopy (XPS). The dissolution of individual $MoO_3$ with acidic PEDOT:PSS was prevented by utilizing an $MoO_3$-PEDOT:PSS conductive composite thin layer and confirmed by analyzing Auger Electron Spectroscopy (AES). The work function of the optimized MoAu/Mo–PPSS conductive composite thin layer electrode was measured at 4.87 eV, which is comparable to the ITO electrode of 4.8 eV [42] using Kelvin Force Microcopy (KFM). We observed that the photoactive layer of P3HT:PCBM absorbed in the UV-Visible wavelength range of 300–650 nm. The device performance of 3.97% based on MoAu/Mo–PPSS conductive composite electrode exhibited comparable enhancement and just 6% enhancement compared to the ITO-based electrode (3.91%). The low work function of the conductive composite, the enhancement of interfacial contact quality between the hole transport layer and the photoactive layer, and the protection of $MoO_3$ dissolution with an acidic PEDOT:PSS layer indicated that the solution-processable Mo–PPSS conductive composite layer on the $MoO_3$/Au layer electrode could substitute the ITO electrodes in the bulk of heterojunction organic photovoltaic cells.

## 2. Experiment

### 2.1. Materials

Ammonium molybdate [((NH$_4$)$_2$MoO$_4$), 99.98% trace metal basis], P3HT [(poly(3-hexylthiophene)), regioregularity $\geq$ 90%], PCBM [(1-(3-methoxycarbonyl)-propyl-1-phenyl-[6,6]C61), >99.9%, MW = 910.88 g/mole], and [o-dichlorobenzene (*o*DCB), anhydrous 99%] were purchased from Sigma–Aldrich, while the PEDOT:PSS (Clevios P, Clevios P VP AI4083, and Clevios PH 1000) were purchased from Heraeus, and Au-evaporating sludge (99.999%) was purchased from iTASCO. The reagents were used without further purification.

### 2.2. Preparation of the Conductive M–PSS Composite

The Mo–PPSS conductive composite was prepared according to the previously published work [43]. The Mo–PPSS conductive composite films were prepared by dissolving 0.5 g of ammonium molybdate (NH$_4$)$_2$MoO$_4$ into 10.0 g of highly conductive PEDOT:PSS (PH1000). During this process, the pH was adjusted up to 7.0 of the mixed solution and the resulting solutions were vigorously stirred for around 1 h at 90 °C.

### 2.3. Device Fabrication

Figure 1 shows an ITO-free organic solar cell with an anodic electrode of MoAu/Mo–PPSS in the MoO$_3$/Au/Mo–PPSS(conductive)/PEDOT:PSS/P3HT:PCBM/LiF/Al device. The glass substrates for OSCs device preparation were cleaned using piranha solution (H$_2$SO$_4$:H$_2$O$_2$ 4:1) for about 10 min, then the glasses were rinsed with deionized (DI) water, and finally dried using N$_2$ blow. The MoO$_3$/Au bi-layers anodic electrodes were deposited onto the clean glass substrates by sequential an electronic beam (e-beam) evaporation using a designed shadow mask to create a patterned electrode on the glass substrate. The substrates were kept at room temperature, while MoO$_3$ and Au were deposited at rates of (0.3 and 0.1) nm/s, respectively, maintaining a base pressure of $2 \times 10^{-6}$ Torr. The thickness and deposition rate of these layers were measured using a quartz crystal microbalance on an electron beam (e-beam) evaporator. Then, the Spin-coated conductive composite thin film was deposited on top of the MoO$_3$/Au layer by spinning at 3000 rpm for 60 s for 10 min, followed by annealing at 120 °C. Spin-coated ca. 65-nm thick PEDOT: PSS (Clevious P) HTL was deposited by spinning at 3000 rpm for 60 s, and annealed with a hot plate at 110 °C for 10 min. A blended photoactive layer of ca. 200-nm thickness of bulk-heterojunction (BHJ) films of P3HT and PCBM in *o*DCB solvent was spin-coated by spinning at 600 rpm for 60 s at a weight ratio of 5:4. Then, the films were kept for 1 h inside the nitrogen-filled glovebox to evaporate the solvent. Subsequently, a 0.3 nm LiF and 150 nm aluminum cathode was thermally deposited through a required designed shadow mask with an active area of 6 mm$^2$ at a base pressure of $2 \times 10^{-7}$ Torr. Finally, the device was completed through thermal post-annealing in a vacuum glove box at 150 °C for 15 min.

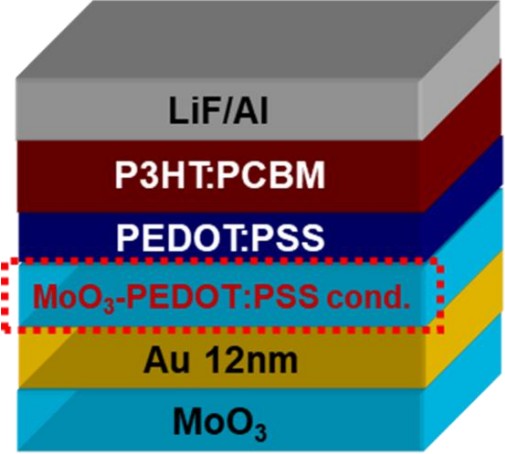

**Figure 1.** Schematic structure of polymer solar cell devices.

In the case of standard ITO/PEDOT:PSS-based device formation, pattern indium-doped tin oxide (ITO), with a thickness of 150 nm coated on glass (13 Ω/sq, FreeM Tech, Hwaseong-si, Republic of Korea), was cleaned in an ultrasonic bath with organic solvents, such as acetone, isopropanol, and was finally cleaned with deionized (DI) water for 10 min each. After drying with $N_2$ blow, an $O_3$/ultraviolet was used on cleaned substrates for 30 min. The next layers of PEDOT:PSS HTL, photoactive layer, LiF, and Al were similarly deposited as described above.

*2.4. Characterization*

Optical and absorption spectra of the resulting thin films were measured using UV-visible spectrophotometry (UV-3150, Shimadzu Corp., Kyoto, Japan). The sheet resistance of the prepared thin films was measured using a four-point probe (Chang-Min Tech Co., Ltd., Yatap, Republic of Korea). To observe the thicknesses and surface morphology of the prepared thin films, field emission scanning electron microscopy (FE–SEM, Jeol-7401F, JEOL Ltd., Tokyo, Japan) was utilized. The surface elemental composition of the composite layer and PEDOT:PSS layer were studied and analyzed using (X-ray photoelectron spectroscopy XPS, K–Alpha, Thermo Fisher Scientific, Massachusetts, United States). The measurements were carried out with an Al source with a photon energy of 1486.6 eV. The depth profiles of the prepared films were carried out by Auger electron spectroscopy (AES), using a physical electronic PHI 680 Auger nanoprobe. The work functions of different prepared electrodes were measured by Kelvin force microscopy (KFM) (PA400, Seiko, Instruments Inc., Chiba, Japan). The resulting current density–voltage (*J–V*) curves of the devices were analyzed under irradiation solar simulator 100 mW/cm$^2$ illumination (AM 1.5 G) in air. Monocrystalline silicon 54 standard solar cells were used to calibrate the illuminated intensity of the light.

**3. Results and Discussion**

Conductive PEDOT:PSS polymer has positively charged PEDOT, and $MoO_3$ has absolute electronegativity. In the composite of $MoO_3$–PEDOT:PSS, Mo in $MoO_3$ and S in PEDOT are bonded, and the resonant configuration of PEDOT chains transforms from a benzoid structure to a quinoid structure, as shown in Figure 2, which progresses the interconnection between the conducting PEDOT units. Therefore, this quinoid structure improves the conductivity of the composite of the $MoO_3$–PEDOT:PSS layer [36]. An electrochromic composite film prepared by $MoO_3$ and PEDOT:PSS both improve electron conduction, and increase ion transport, as well as ion diffusion by modifying the cumulative configuration of PEDOT. From our previous reports, this matches well with the existence of the $Mo^{5+}$ valence electronic state formation from $Mo^{6+}$ linked with the oxidation state of PEDOT [43], which suggests that there is a chemical reaction between the $MoO_3$ and PEDOT chains. Meanwhile, charge transfer occurs between the $MoO_3$ and PEDOT chains. The plausible reactions are shown below:

$$PEDOT : PSS + MoO_3 \left( (NH_4)_2 MoO_4 \right) \rightarrow PEDOT + PSS + MoO_x^-; \ x = (2 \ to \ 3)$$

**Figure 2.** Benzoid and quinoid form of PEDOT [44] in the presence of $MoO_3$.

PEDOT:PSS deposited onto various substrates has different interfacial contact qualities to facilitate charge transfer at the interface of the electrode. Polymer materials, as a hole transport layer of PEDOT:PSS, can be easily spin-coated onto the MoAu-(Mo–PPSS)

conductive electrode. The polymeric HTL PEDOT:PSS has good interaction with the composite of the Mo–PPSS layer on the $MoO_3$-Au electrode, as both the layer contains aromatic structures, i.e., benzoid and quinoid. This creates a smooth surface, leading to an improved interfacial contact quality and thereby facilitating charge transfer at the electrode interface, which can be explained by the deconvoluted results of the XPS analysis as shown in Figures 3 and 4.

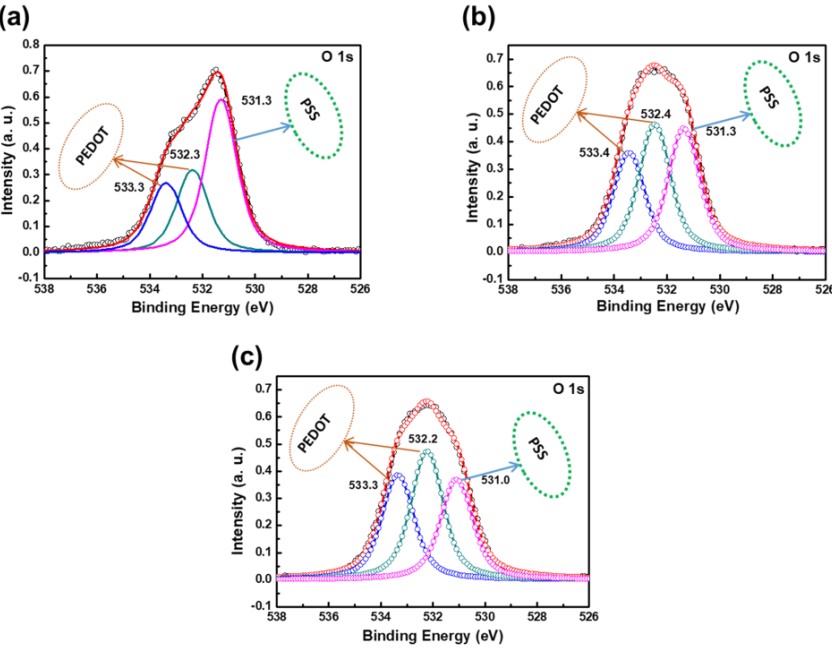

**Figure 3.** XPS profile of the core level O 1S for PEDOT:PSS on (**a**) MoAuMo (**b**) MoAu/Mo–PPSS composite and (**c**) MoAu/Mo–PPSS conductive composite.

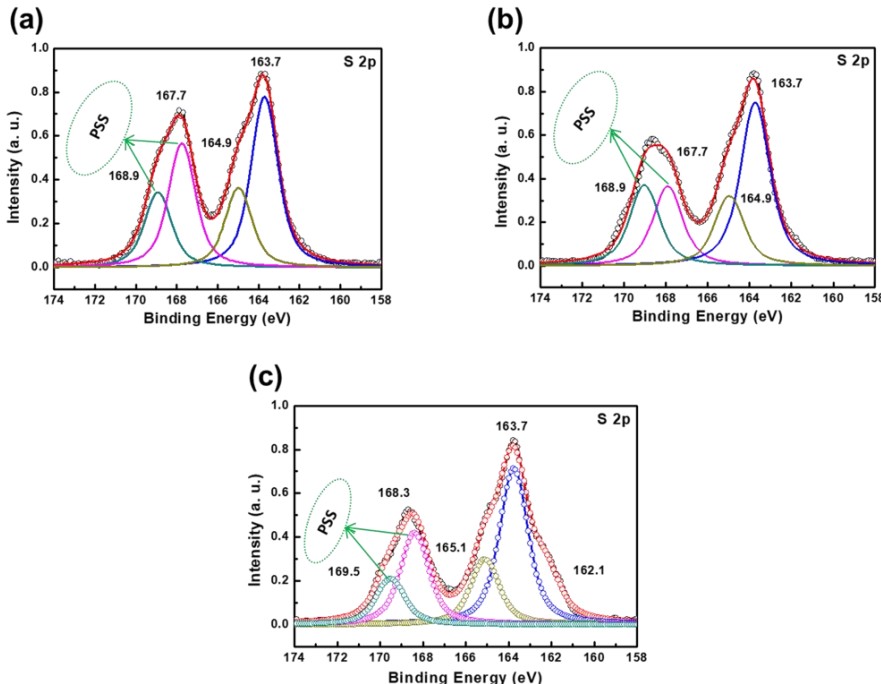

**Figure 4.** XPS data of the core level S 2p for PEDOT:PSS on ((**a**) MoAuMo (**b**) MoAu-(Mo–PPSS) composite and (**c**) MoAu-(Mo–PPSS) conductive composite.

An elemental surface composition profile of the PEDOT:PSS film onto different electrodes was investigated by utilizing XPS, as displayed in Figure 3. Figure 3a–c shows the raw peak, fitting peak, and the deconvoluted peak of the core level O 1s for PEDOT:PSS on MoAuMo, MoAu/Mo–PEDOT:PSS, and MoAu-(Mo–PPSS) conductive composite, respectively. All the spectra disclose three different intensity major peaks at a binding energy of 531 eV, which is well known for the O–S of PSS moiety, and both (532 and 533) eV are well known for the C–O–C bond of PEDOT moiety in the case of PEDOT:PSS, respectively [42]. Comparatively, the surface PEDOT:PSS in different electrodes shows different intensities of deconvoluted peaks. The MoAu/Mo–PPSS conductive composite electrode shows an intense PEDOT peak and a reduced PSS peak, compared to the other electrodes. This suggests that the alteration of electronic charge distribution along with the C–O–C bond of the PEDOT occurred, due to the effect of the different interfaces on the different electrodes and the structural transformation from a benzoid to a quinoid form. As the PSS moiety is reduced, the acidity of the resulting composite is also reduced. Which leads to a more stable interface with the electrode.

Figure 4a–c shows the raw peak, fitting peak, and the deconvoluted peak of the S 2p core level for the PEDOT:PSS on MoAuMo, MoAu-(Mo–PPSS) composite, and MoAu/Mo–PPSS conductive composite, respectively. The S 2p peak shows S 2p3/2 and S 2p1/2 spin-split doublet with 1.2 eV splitting energy, and thus these two peaks have a 1:2 intensity ratio [45,46]. The peaks involved at (163.7 and 164.7) eV of the S 2p contribution peaks belong to sulfur (S) atoms of the PEDOT, while the high binding energies peaks at (167.7 and 168.9) eV correspond to PSS. The PSS shows a higher binding energy than the PEDOT, due to the high electronegative oxygen attached to the S atom in the sulfonate fragment of the PSS. On the other hand, in Figure 4c, the intensity of the S 2P from the PSS moiety contribution remarkably decreases in contrast to the contribution from PEDOT, which increases compared to the other PEDOT:PSS on different electrodes. This indicates the quality control electrical contact film formed for PEDOT:PSS on the MoAu-(Mo–PPSS) conductive composite [47]. It has been recognized that the hygroscopic and highly acidic nature of PSS can easily absorb water, which results in the serious degradation of conductivity, as well as the stability of organic optoelectronics with PEDOT:PSS electrode devices [48,49]. The removal of insulating PSS from the upper part of the surface of the PEDOT:PSS grains and the crystallization of PEDOT lead to the establishment of highly conductive PEDOT-rich giant grains that enhance the transportation of charge carriers, as well as increase the photovoltaic device performance of the P3HT:PCBM photoactive layer cells by modifying the PEDOT:PSS layer [50].

The thin layer of $MoO_3$ is often soluble in water, and when the acidic water-dispersed PEDOT:PSS spin coats onto the MoAuMo electrode, as shown in Figure 5a,b the water-dispersed PEDOT:PSS washes away to dissolve top $MoO_3$ of MoAuMo structure. As a result, the AES analysis results showed that only the upper portion of the Au-coated layer had a nearly zero percent (atomic concentration) of $MoO_3$ remaining, as is stated in a prior article [43]. As shown in Figure 5b, the pH-neutral $MoO_3$–PEDOT:PSS conductive composite that was spin coated between MA and the acidic PEDOT:PSS (HTL) prevented $MoO_3$ from dissolving in the MoAu-(MoPPSS) conductive composite. As a result, as can be seen in Figure 5b, some $MoO_3$ was still present on top of the Au film shown by the AES analysis. The remaining $MoO_3$ can produce $Mo^{6+}$ to $Mo^{5+}$ with a tiny amount of vacancy oxygen ($MoO_3^-$). The resulting oxygen vacancies arise due to the transfer of Hx from the acidic PEDOT:PSS to $MoO_3$, which tends to form Hx $MoO_3$ when the conductive composite of $MoO_3$–PEDOT:PSS was prepared [28,51]. The transformation of $Mo^{6+}$ to $Mo^{5+}$ helps to decrease the work function of the composite, which is shown in Figure 6.

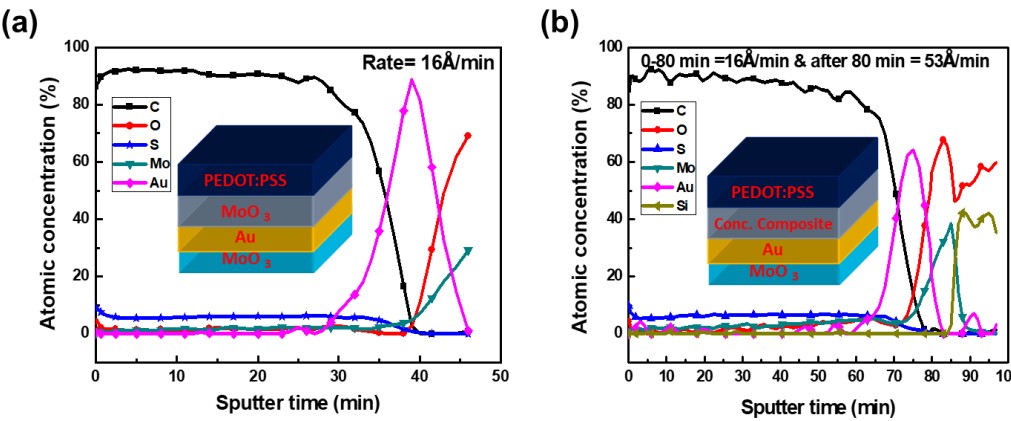

**Figure 5.** AES profile of PEDOT:PSS on (**a**) MoAuMo layer and (**b**) MoAu-PPSS conductive composite layer.

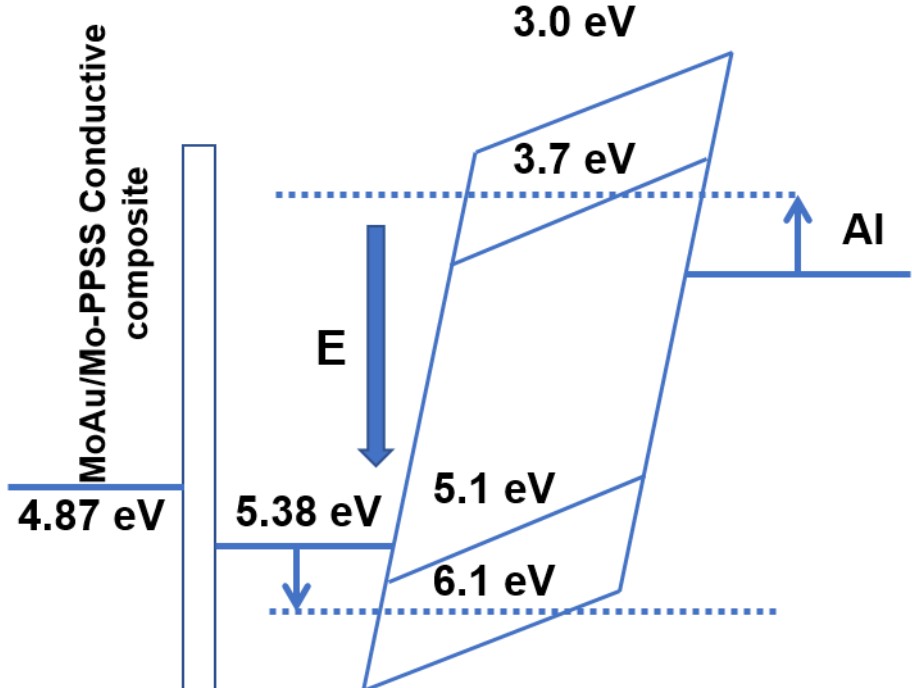

**Figure 6.** Schematic energy band diagram of experimental organic solar cells.

The work function is the representation of the energetic requirements for adding or withdrawing an electron from or to a solid for charge exchange. Hence, different electrodes' work function were measured by utilizing Kelvin force microscopy (KFM) (Seiko, PA400, Japan). The conductive MoAu-(Mo–PPSS) composite electrode work function was measured at 4.87 eV, which is comparable with the reported ITO electrode value of 4.8 eV, as shown in Figure 6. The minimum adjustment of active components energy levels is essential for the MoAu-(Mo–PPSS) composite electrode in the replacement of the ITO anode in photovoltaic devices. Generally, low cation oxidation states have the effect of reducing the work function of an oxide [52]. In the case of the composite formation of $Mo^{6+}$ being reduced to $Mo^{5+}$, that is why the composite electrode WF decreases to 4.87 eV. Moreover, the WF of PEDOT:PSS HTL onto the composite was 5.38 eV, which aligns with that of the anode and the highest occupied orbital (HOMO) energy level of the absorber P3HT:PCBM, and hence, enhances the extraction of holes from the photoactive layer to the anode. This higher rate of charge transfer leads to lower recombination rates. Hence, overall device performance improves.

High transparency is an important prerequisite for the function of solar cells, which depends on photoactive layer absorption. Hence, to create an efficient organic solar cell, an anode with high transparency is essential. To characterize the optical characteristics of the ITO-free conductive MoAu-(Mo–PPSS) composite electrode, the transmittances of the conductive MoAu-(Mo–PPSS) composite electrode with various thicknesses of the conductive composite film were measured utilizing UV-Visible spectrophotometry, as displayed in Figure 7a. The MoAu/Mo–PPSS (3000 rpm) electrode exhibits excellent transparency (~78%) in the full spectrum range, as shown in Figure 7a. In the wavelength range of 500–650 nm, which is the main absorption region of P3HT, the transmission of both the MoAu-(Mo–PPSS) (3000 rpm) electrode and MoAuMo electrode were similar. The transmission of the Mo–PPSS film gradually decreases with thickness. Therefore, we optimized the thickness of 54 nm for the MoAu/Mo–PPSS conductive composite film by using FE-SEM for the consequent device formation. Conductivity was observed for the 54-nm thick MoAu/Mo–PPSS conductive composite, as well as the 150-nm thick commercial ITO electrode. The conductivity of electrodes was calculated from the sheet resistance and the thickness using the simple equation as follows:

$$\text{Conductivity} = \frac{1}{Resistivity} = \frac{1}{R_s \cdot t} \tag{1}$$

where $R_s$ and $t$ are the sheet resistance of the film measured using a four-point probe and the thickness of the film, respectively. The sheet resistance of 13 ohm/sq. for the commercial ITO electrode was measured using a four-point probe, and the conductivity of 512 S/cm was calculated for 150 nm ITO using Equation (1). The conductivity of the ITO electrode is still 2.70 times larger compared to the conductive MoAu-(Mo–PPSS) composite electrode. However, the reasonable conductivity of 190 S/cm and excellent transparency (~78%) in the full spectrum range of MoAu/Mo–PPSS conductive electrode showed suitable replacement with ITO electrode. The 54 nm thick MoAu/Mo–PPSS conductive composite film showed comparable conductivity and transmittance in comparison to the ITO electrode. Determining the absorbance of the photoactive layer is an essential criterion for photovoltaics. Figure 7b shows UV-Vis absorption spectra of the P3HT:PCBM photoactive layer on the different electrodes. The photoactive layer absorbs in the range of 300–650 nm and the maximum absorption is observed at 515 nm (2.40 eV). There is a local peak at 335 nm, influenced by the PCBM in the blend with P3HT, and three other peaks at 515 nm, 550 nm, and 602 nm, due to the appearance of P3HT. These peaks were in the same absorption ranges and were consistent with published data [53,54]. Maximum absorption was observed at 515 nm wavelength for ITO/PEDOT:PSS electrode and comparatively less absorption was observed for MoAu/Mo–PPSS in the same wavelength. MoAuMo/PEDOT:PSS showed the lowest absorption as PEDOT:PSS deteriorates the MoO$_3$ layer on the MoO$_3$/Au electrode.

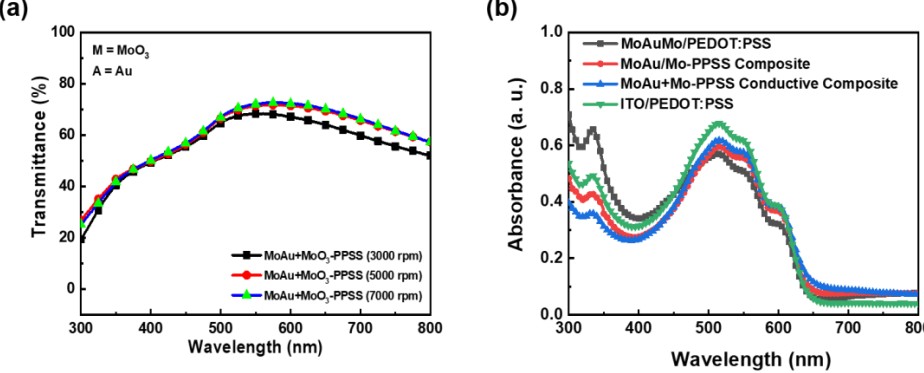

**Figure 7.** (**a**) Optical transmittances of Glass/MoO$_3$ (30 nm)/Au (12 nm)/Mo–PPSS conductive composite, and (**b**) UV-Visible absorption spectra of P3HT:PCBM photoactive layer on different electrodes.

Figure 8 presents the current-voltage characteristics of the devices under 1.5 G irradiation (100 mW/cm$^2$) using different electrodes, while Table 1 tabulates the calculated device parameters. Both the shunt and the series resistances ($R_{sh}$ and $R_s$) were calculated using the inverse slope values of the *J–V* curves at a voltage of (0.0 and 1.5) V, respectively [55]. The highest $R_s$ of (22.21 $\pm$ 1.66 $\Omega$ cm$^2$) with the lowest $R_{sh}$ of (18.25 $\pm$ 0.95 $\Omega$ cm$^2$) were obtained for the MoAuMo/PEDOT:PSS composite electrode-based device. On the other hand, the lowest $R_s$ of (11.70 $\pm$ 0.57 $\Omega$ cm$^2$) with acceptable $R_{sh}$ of (313.00 $\pm$ 0.63 $\Omega$ cm$^2$) were obtained for the MoAu/Mo–PPSS conductive composite electrode-based device. The highest current density ($J_{sc}$) of 12.47 $\pm$ 0.18 mA/cm$^2$ and highest $R_{sh}$ of (576.00 $\pm$ 0.50 $\Omega$ cm$^2$) were observed for the ITO/PEDOT:PSS-based device. The fabricated devices with ITO/PEDOT:PSS and ITO-free MoAuMo/PEDOT:PSS exhibited a Power Conversion Efficiency (PCE) of (2.89 and 1.84)%, respectively. The low efficiency of the MoAuMo/PEDOT:PSS was due to the removal of the top MoO$_3$ by the reaction between the acidic and hygroscopic PEDOT:PSS and MoO$_3$ [24], and the low $J_{sc}$ was due to less absorption in the photoactive P3HT:PCBM layer on the MoAuMo/PEDOT:PSS electrode, as shown in Figure 7b. In addition, the fill factor (FF) was reduced due to the high series resistance, which also significantly impacted the reduction in the current density. On the other hand, the MoAu/Mo–PPSS conductive composite electrode-based device exhibited a PCE of 3.97%, with a high $V_{oc}$ of 0.64 V compared to the ITO electrode-based device ($V_{oc}$ = 0.60 V). The enhancement of $V_{oc}$ for the MoAu/Mo–PPSS conductive composite electrode might be due to the tuning of the WF of HTLs, which was well matched with the deep HOMO-level small molecule donor materials [56]. The highest standard deviation (SD) of $J_{sc}$ ($\pm$0.49 mA/cm$^2$), $R_{sh}$ ($\pm$0.95 $\Omega$ cm$^2$), and $R_s$ ($\pm$1.66 $\Omega$ cm$^2$) was obtained for the MoAuMo/PEDOT:PSS electrode-based device. This is because the corrosion of the MoO$_3$ thin film by acidic as well as hygroscopic PEDOT:PSS is uncontrolled. Whereas the SD of MoAu/M-PSS electrode-based devices are less and similar to the ITO electrode-based devices, as conductive Mo–PPSS and PEDOT:PSS HTL layers form stable, smooth, and favorable interfaces.

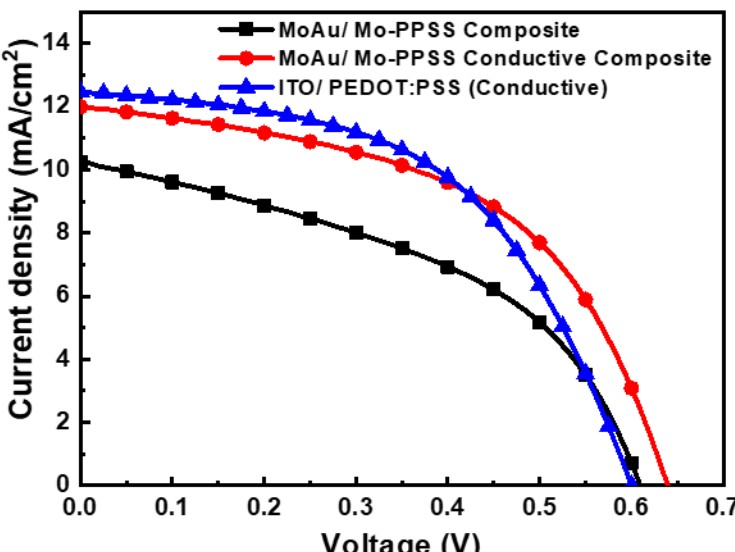

**Figure 8.** Current-voltage (*J–V*) characteristics of organic solar cells fabricated with different electrodes.

**Table 1.** Photovoltaic device performance parameters for ITO-based and ITO-free $MoO_3/Au/Mo$–PPSS composites with different conductivity-based electrodes. SD denotes the standard deviation and SD calculated from four devices.

| | $J_{sc}$ mA/cm$^2$ | $V_{OC}$ V | FF % | H % | $R_{sh}$ $\Omega$ cm$^2$ | $R_s$ $\Omega$ cm$^2$ |
|---|---|---|---|---|---|---|
| ITO/PEDOT:PSS | $12.47 \pm 0.18$ | $0.60 \pm 0.01$ | $0.52 \pm 0.01$ | $3.91 \pm 0.02$ | $576.00 \pm 50$ | $12.86 \pm 0.44$ |
| MoAuMo/PEDOT:PSS | $7.75 \pm 0.49$ | $0.55 \pm 0.02$ | $0.43 \pm 0.02$ | $1.84 \pm 0.05$ | $18.25 \pm 95$ | $22.21 \pm 1.66$ |
| MoAu/M-PSS Composite | $10.25 \pm 0.39$ | $0.61 \pm 0.01$ | $0.45 \pm 0.01$ | $2.81 \pm 0.10$ | $165.57 \pm 75$ | $13.48 \pm 0.84$ |
| MoAu/Mo–PPSS Conductive Composite | $11.99 \pm 0.19$ | $0.64 \pm 0.01$ | $0.52 \pm 0.01$ | $3.97 \pm 0.03$ | $313.00 \pm 63$ | $11.70 \pm 0.57$ |

## 4. Conclusions

In conclusion, the ITO electrode was replaced by using solution-process spin-coated $MoO_3$–PEDOT:PSS conductive composite on $MoO_3/Au$ multilayers, and their optical as well as electrical properties were investigated. An elemental composition of the hole transport PEDOT:PSS layer was investigated using XPS analysis, and the effects of water dispersal and acidic PEDOT:PSS were examined by AES analysis. A conductive composite $MoO_3/Au/MoO_3$–PPSS electrode work function of 4.87 eV was measured using KFM, which was similar to the reported ITO electrode of 4.8 eV [40]. From UV-Visible spectroscopy measurements, we observed that the P3HT:PCBM photoactive layer absorbs in the UV-Vis wavelength range of 300–650 nm, and maximum absorption was observed at 515 nm. A PCE of 3.97% based on the $MoO_3/Au/Mo$–PPSS conductive composite electrode presented a comparable 6% enhancement compared to the ITO-based electrode (3.91%). The enhancement of PCE based on the conductive composite electrode was mainly due to the high conductivity, interfacial contact quality, and low work function of the conductive layer. Thus, this conductive and solution-processed-based multilayer composite electrode was determined to be a promising replacement electrode for ITO-free organic optoelectronic devices.

**Author Contributions:** Conceptualization, methodology, software, M.M. and R.A.; validation, formal analysis, and investigation, M.M., R.A. and M.A.K.S.; data curation, writing—review and editing, M.M., R.A, M.A.K.S. and S.S.; writing—original draft preparation, M.M., R.A. and M.A.K.S.; visualization, writing—review and editing, and supervision, J.L. All authors have read and agreed to the published version of the manuscript.

**Funding:** This study was performed with a research grant from the National Research Foundation of Korea (NRF) sponsored by the Ministry of Science and ICT of the Republic of Korea (grant number: 2022R1A5A7000765).

**Data Availability Statement:** All data associated with the study are available in the manuscript.

**Conflicts of Interest:** Regarding the publishing of this work, the authors state that they have no conflict of interest.

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
