# Peer review of "Conductive MoO3–PEDOT:PSS Composite Layer in MoO3/Au/MoO3–PEDOT:PSS Multilayer Electrode in ITO-Free Organic Solar Cells"

_processes, doi:10.3390/pr11020594_

Round 1

Reviewer 1 Report

Recommendation

Minor Revision

Comments:

This manuscript reports the utilization of an MoO3–PEDOT:PSS composite layer in the ITO-free organic solar cell. The solar cells based on the MoO3/Au/M−PSS conductive composite exhibit an improvement of PCE of 3.97 %, comparable to that of the ITO-based solar cells (3.91 %). The enhancement of PCE was mainly due to the high conductivity, low work function of the conductive composite, and enhancement of interfacial contact quality between the hole transport layer and the photoactive layer. The solution-processing method for preparing the hole transport layer is facile and effective. Some minor issues that need to be tackled are as follows.

1. Why is the Jsc of the solar cells based on the MA-M-PSS composite layer lower than that of the ITO-based solar cells?

2. All the parameters in Table 1 should be given with standard deviations.

3. On the top of page 4: Please check the structural drawings of the quinoid structure in the Scheme. From my perspective, it is just a cationic radical form of a benzoid structure. Moreover, the PEDOT polymer is not fully oxidized by MoO3. Therefore, more building units are needed to illustrate the quinoid structure clearly (please see a recent review: Chem. Rev. 2022, 122, 4325–4355).

4. In section 2.1 Materials on page 2: “ODCB” should be “oDCB”.

Author Response

We would like to thank the reviewers for their time and valuable comments on our manuscript entitled "Conductive MoO3–PEDOT:PSS composite layer in MoO3/Au/MoO3–PEDOT:PSS multilayer electrode in ITO-free organic solar cells” (Manuscript ID: processes-2188974). We appreciated it as all these comments were precious and helpful for improving the quality of this manuscript. We have carefully studied their feedback and have revised our manuscript to address all the comments and suggestions. We hope this revision will meet your approval to attain publication in the journal of processes.

Our detailed response to the reviewers’ comments is as follows (modification in the revised manuscript is indicated by green highlight):

REVIEWER 1

This manuscript reports the utilization of an MoO3–PEDOT:PSS composite layer in the ITO-free organic solar cell. The solar cells based on the MoO3/Au/M−PSS conductive composite exhibit an improvement of PCE of 3.97 %, comparable to that of the ITO-based solar cells (3.91 %). The enhancement of PCE was mainly due to the high conductivity, low work function of the conductive composite, and enhancement of interfacial contact quality between the hole transport layer and the photoactive layer. The solution-processing method for preparing the hole transport layer is facile and effective. Some minor issues that need to be tackled are as follows.

Response:

Thank you very much for your review.

Detailed comments and their responses are listed below:

Comment 1:

Why is the Jsc of the solar cells based on the MA-M-PSS composite layer lower than that of the ITO-based solar cells?

Response:

Thank you very much for your concern. The low conductivity of MA-M-PSS composite layer is the main reason. The current flow across the device is directly related to the conductivity of the electrodes, along with other parameters. Other possible reasons are the high interfacial resistance of the MA-M-PSS and less absorption of the photoactive layer on the MA-M-PSS composite electrode due to less transmittance. Due to low conductivity, less transmittance, and high interfacial resistance, the series resistance increased and as a result, Jsc decreased. However, both conductivity and interfacial contact were improved by using conductive composite and MA+M-PSS conductive composite shows less difference of  Jsc (< 0.5 mA/cm2) with ITO.

Comment 2:

All the parameters in Table 1 should be given with standard deviations.

Response:

Thank you very much for your suggestion. Based on your suggestion, the revised Table 1 has been modified as follows:

Prevised manuscript Table 1:

Table 1. Photovoltaic performance parameters for ITO-based and ITO-free MoO3/Au/M-PSS composite-based devices with different conductivity.

Jsc
mA/cm2

VOC
V

FF
%

η
%

Rsh
Ω cm2

Rs
Ω cm2

ITO/PEDOT: PSS

12.47

0.60

0.52

3.91

576.00

12.86

MAM/PEDOT: PSS

7.75

0.55

0.43

1.84

18.25

22.21

MA/ M-PSS Composite

10.25

0.61

0.45

2.81

165.57

13.48

MA+M-PSS Conductive Composite

11.99

0.64

0.52

3.97

313.00

11.70

Revised manuscript Table 1:

Table 1. Photovoltaic performance parameters for ITO-based and ITO-free MoO3/Au/M-PSS composite-based devices with different conductivity. SD denotes the standard deviation and SD calculated from 4 devices.

Jsc
mA/cm2

VOC
V

FF
%

η
%

Rsh
Ω cm2

Rs
Ω cm2

ITO/PEDOT: PSS

12.47±0.18

0.60±0.01

0.52±0.01

3.91±0.02

576.00±50

12.86±0.44

MAM/PEDOT: PSS

7.75±0.49

0.55±0.02

0.43±0.02

1.84±0.05

18.25±95

22.21±1.66

MA/ M-PSS

Composite

10.25±0.39

0.61±0.01

0.45±0.01

2.81±0.10

165.57±75

13.48±0.84

MA+M-PSS

Conductive Composite

11.99±0.19

0.64±0.01

0.52±0.01

3.97±0.03

313.00±63

11.70±0.57

Comment 3:

On the top of page 4: Please check the structural drawings of the quinoid structure in the Scheme. From my perspective, it is just a cationic radical form of a benzoid structure. Moreover, the PEDOT polymer is not fully oxidized by MoO3. Therefore, more building units are needed to illustrate the quinoid structure clearly (please see a recent review: Chem. Rev. 2022122, 4325–4355).

Response:

Thank you very much for your valuable suggestion. We agree that your suggestion is correct. We read your suggested review paper which illustrates the quinoid structure clearly. Based on your suggestion the revised benzoid and quinoid structure of PEDOT have been modified as a Figure 2. And the reference is cited in the explanation section.

Prevised manuscript structural drawing:

Revised manuscript structural drawing:

Figure 2. Benzoid and quinoid form of PEDOT in presence of MoO3.

Reference 45 has been added in the revised manuscript: N. A. Kukhta et al. Chem. Rev. 2022, 122, 4, 4325-4355.

Comment 4:

In section 2.1 Materials on page 2: “ODCB” should be “oDCB”.

Response:

We apologize for our mistake. Thank you very much for your correction. Based on your correction the revised manuscript has been modified as follow:

Prevised manuscript section 2.1 Materials on page 2: o-dichlorobenzene (ODCB)

Revised manuscript section 2.1 Materials on page 2: o-dichlorobenzene (oDCB)

Reviewer 2 Report

Authors prepared MoO3–PEDOT:PSS conductive composite (M−PSS conductive), which was used to fabricate ITO-free organic solar cells. This manuscript is well prepared, but the novelty is not high enough. Therefore, I suggest accepting this paper after a major revision. Before resubmitting, the authors should solve the following problems:

1. The abstract part is too simple, the novelty of this work should be clearly mentioned. In addition, the photoactive materials for solar cells should be mentioned.

2. “The solar cells based on the MoO3/Au/M−PSS conductive compo-site exhibit an improvement of power conversion efficiency (PCE) of 3.97 %, comparable to that of the ITO-based solar cells PCE of 3.91 %.” The expression is not appropriate, the error may be bigger than improvement.

3. The introduction part should be carefully revised to emphasize the novelty of this work. In addition, the author should discuss that why chose P3HT/PCBM as photoactive materials, because non-fullerene solar cells are more popular right now. The following references relevant to P3HT/PCBM should be included for more readable, Dyes and Pigments, 2018, 158, 213–218; Frontiers in Energy Research, 2018, 6, 113.

4. The authors should offer more information about the materials. For example the molecular weight and stereoregularity of P3HT, these information is very important for the device performances.

5. The authors should offer fabrication detail of devices with ITO.   

6. The figures are too vague to see clearly.

7. In the abstract part, “The enhancement of PCE was mainly due to the high conductivity, low work function of the conductive composite, and enhancement of interfacial contact quality between the hole transport layer and the photoactive layer” . Except for work function, other factors were not discussed in the manuscript. Also, surface morphology was not discussed, but was mentioned in the introduction part.

8. In order to disclose the reason of improvement, at least, the UV and EQE of devices should be provided.

Author Response

We would like to thank the reviewers for their time and valuable comments on our manuscript entitled "Conductive MoO3–PEDOT:PSS composite layer in MoO3/Au/MoO3–PEDOT:PSS multilayer electrode in ITO-free organic solar cells” (Manuscript ID: processes-2188974). We appreciated it as all these comments were precious and helpful for improving the quality of this manuscript. We have carefully studied their feedback and have revised our manuscript to address all the comments and suggestions. We hope this revision will meet your approval to attain publication in the journal of processes.

Our detailed response to the reviewers’ comments is as follows (modification in the revised manuscript is indicated by green highlight):

REVIEWER 2

Comments and Suggestions for Authors:

Authors prepared MoO3–PEDOT:PSS conductive composite (M−PSS conductive), which was used to fabricate ITO-free organic solar cells. This manuscript is well prepared, but the novelty is not high enough. Therefore, I suggest accepting this paper after a major revision. Before resubmitting, the authors should solve the following problems:

Response:

Thank you very much for your review and suggestion. We corrected and tried our best to solve your suggested problems.

Comment 1:

The abstract part is too simple, the novelty of this work should be clearly mentioned. In addition, the photoactive materials for solar cells should be mentioned.

Response:

Thank you very much for your comment. According to your suggestion, the abstract is rewritten and this work's novelty is included in the revised manuscript. In addition, the photoactive materials for solar cells have been mentioned in the revised manuscript.

Prevised manuscript abstract: The solution-processable and conductive MoO3–PEDOT:PSS composite layer in MoO3/Au/MoO3–PEDOT:PSS multilayer electrode in the ITO-free organic solar cell was optimized in terms of electrical conductivity, interfacial contact quality, work function, and process wettability of the conductive composite film. The solar cells based on the MoO3/Au/M−PSS conductive composite exhibit an improvement of power conversion efficiency (PCE) of 3.97 %, comparable to that of the ITO-based solar cells PCE of 3.91 %. The enhancement of PCE was mainly due to the high conductivity, low work function of the conductive composite, and enhancement of interfacial contact quality between the hole transport layer and the photoactive layer. These results indicate that the solution-processable MoO3–PEDOT:PSS conductive composite on MoO3/Au multilayer electrode can replace the ITO-based electrode in the bulk heterojunction organic photovoltaics.

Revised manuscript abstract: The solution-processable and conductive MoO3–PEDOT:PSS composite layer in MoO3/Au/MoO3–PEDOT:PSS multilayer electrode in the ITO-free organic solar cell was optimized in terms of electrical conductivity, interfacial contact quality, work function, and process wettability of the conductive composite film. The surface composition of the PEDOT:PSS film onto different electrodes was observed by using X-Ray Photoelectron Spectroscopy. The PEDOT:PSS-MoO3 composite protects the dissolution of individual MoO3 with PEDOT:PSS, which was confirmed by Auger Electron Spectroscopy. The UV-Visible spectroscopy showed that the photoactive layer of P3HT:PCBM absorbs in the wavelength range of 300-650 nm with the maximum absorption at 515 nm (2.40 eV). The device performance of 3.97 % based on MoO3/Au/M−PSS conductive composite electrode exhibit comparable and just 6% enhancement compared to ITO-based electrode (3.91 %). The enhancement of device efficiency was mainly due to the high conductivity, the low work function of the conductive composite, and the enhancement of interfacial contact quality between the hole transport layer and the photoactive layer. These results indicate that the solution-processable MoO3–PEDOT:PSS conductive composite on MoO3/Au multilayer electrode can replace the ITO-based electrode in the bulk heterojunction organic photovoltaics.

Comment 2:

“The solar cells based on the MoO3/Au/M−PSS conductive compo-site exhibit an improvement of power conversion efficiency (PCE) of 3.97 %, comparable to that of the ITO-based solar cells PCE of 3.91 %.” The expression is not appropriate, the error may be bigger than improvement.

Response:

Thank you very much for your review. We agree with your comment and modified the sentence in the revised manuscript as follows: “The device performance of 3.97 % based on MoO3/Au/M−PSS conductive composite electrode exhibit comparable and just 6% enhancement compared to ITO-based electrode (3.91 %).

Comment 3:

The introduction part should be carefully revised to emphasize the novelty of this work. In addition, the author should discuss that why chose P3HT/PCBM as photoactive materials, because non-fullerene solar cells are more popular right now. The following references relevant to P3HT/PCBM should be included for more readable, Dyes and Pigments, 2018, 158, 213–218; Frontiers in Energy Research, 2018, 6, 113.

Response:

Thank you very much for your suggestions and remarks. We agree with your suggestion that non-fullerene solar cells are more popular right now. Regioregular poly (3-hexylthiophene) (P3HT) has an extremely important structural, photophysical, laminar crystal structure that enables high hole mobility (>10-2 cm2 /Vs) [40], despite the mismatch of its absorption coefficient with the solar emission spectrum. To minimize the mismatch of P3HT absorption, PC61BM is commonly used due to its ball-like fully conjugated structures which have strong electron affinity and unipolar electron transport that promotes the delocalization of electrons [41]. So, combined P3HT/PCBM as photoactive materials lead to devices with high PCE.

References 38 and 39 have been added in the revised manuscript relevant to P3HT/PCBM photoactive layer. Reference 39 is corrected as Frontiers in Energy Research, 2021, 6, 640664. instead of Frontiers in Energy Research, 2018, 6, 113. as you suggested.

[38] Y. Wang et al. Dyes and Pigments, 2018, 158, 213–218.

[39] F. Otieno et al. Frontiers in Energy Research, 2021, 9, 640664.

[40] Y. Qin et al. Adv. Mater. 2016, 28, 9416–9422.

[41] S. Gélinas, et al. (2014). Science 343, 2014, 512–516.

Comment 4:

The authors should offer more information about the materials. For example the molecular weight and stereoregularity of P3HT, these information is very important for the device performances.

Response:

Thank you very much for your vital information. More information about the materials in the revised manuscript has been included as follows: Ammonium molybdate [((NH4)2MoO4), 99.98 % trace metal basis], Au-evaporating sludge, P3HT [(poly(3-hexylthiophene)), regioregularity ≥ 90%], PCBM [(1-(3-methoxycarbonyl)-propyl-1-phenyl-[6,6]C61), >99.9 %, MW= 910.88 g/mole], and [o-dichlorobenzene (oDCB), anhydrous 99%] were purchased from Sigma–Aldrich,

Comment 5:

The authors should offer fabrication detail of devices with ITO.   

Response:

Thank you very much for your suggestion. Fabrication detail of devices with ITO has been included in the revised manuscript as follows: Pattern indium doped tin oxide (ITO) with a thickness of 150 nm coated on glass (13 Ω/sq, FreeM Tech) was cleaned in an ultrasonic bath in acetone, isopropanol and deionized (DI) water for each of 10 min. An O3/ultraviolet was used for 30 min on cleaned substrates after drying with N2 blowing gas. The next layer of PEDOT:PSS, photoactive layer, LiF, and Al were deposited similarly as described above.

Comment 6:

The figures are too vague to see clearly.

Response:

Thank you very much for your comment. We apologize for the vague figure. The figures are modified in the revised manuscript to see clearly.

Comment 7:

In the abstract part, “The enhancement of PCE was mainly due to the high conductivity, low work function of the conductive composite, and enhancement of interfacial contact quality between the hole transport layer and the photoactive layer” . Except for work function, other factors were not discussed in the manuscript. Also, surface morphology was not discussed, but was mentioned in the introduction part.

Response:

Thank you very much for your comments. We agree with your comments. Conductivity and interfacial contact between the hole transport layer and the photoactive layer are discussed in the abstract as well as result and discussion section. In addition, surface morphology term has been removed from the introduction part.

Comment 8:

In order to disclose the reason of improvement, at least, the UV and EQE of devices should be provided.

Response:

Thank you very much for your comment. We agree with your suggestion that at least, the UV and EQE of devices should provide. We include V-visible absorption spectra of P3HT:PCBM photoactive layer as in Fig 7 (b). But due to a lack of EQE facilities and the short time we did not measure the EQE and we apologize for that.

Reviewer 3 Report

The novelty of this work is not clearly given, as the authors had already published similar works in previous years. 

Author Response

We would like to thank the reviewers for their time and valuable comments on our manuscript entitled "Conductive MoO3–PEDOT:PSS composite layer in MoO3/Au/MoO3–PEDOT:PSS multilayer electrode in ITO-free organic solar cells” (Manuscript ID: processes-2188974). We appreciated it as all these comments were precious and helpful for improving the quality of this manuscript. We have carefully studied their feedback and have revised our manuscript to address all the comments and suggestions. We hope this revision will meet your approval to attain publication in the journal of processes.

Our detailed response to the reviewers’ comments is as follows (modification in the revised manuscript is indicated by green highlight):

REVIEWER 3

Comments and Suggestions for Authors:

The novelty of this work is not clearly given, as the authors had already published similar works in previous years.

Response:

Thank you very much for your review. We agree with your comments. The novelty of this manuscript includes

  1. ITO electrode was replaced by using triple layer electrode (MoO3/Au/MoO3-PEDOT:PSS conductive composite) in organic solar cells.
  2. Hygroscopic MoO3 dissolution with acidic PEDOT:PSS was prevented by utilizing composite and confirmed by analyzing Auger Electron Spectroscopy (AES).
  3. The surface composition of the PEDOT:PSS film onto different electrodes was investigated by utilizing X-ray Photoelectron Spectroscopy (XPS) analysis.
  4. The work function of the MoO3/Au/MoO3−PSS conductive composite electrode was measured at 4.87 eV by using Kelvin Force Microscopy (KFM), which is comparable to the ITO electrode of 4.8 eV.
  5. Utilizing solution process MoO3 with conductive PEDOT:PSS composite electrode enhances the conductivity as well as device efficiency of 6% compared to ITO electrode-based devices.

We did some work on the replacement of ITO electrodes by utilizing different electrodes including interlayers. The differences are summarized in the table as follows:

Electrode

Jsc
mA/cm2

VOC
V

FF
%

η
%

Ref.

ITO/PEDOT: PSS

12.47

0.60

0.52

3.91

This work

MoO3/Au/(MoO3-PSS conductive composite)/ PEDOT:PSS

11.99

0.64

0.52

3.97

This work

ITO

7.80

0.67

0.48

2.50

previous work13

PEDOT:PTS

4.90

0.68

0.42

1.40

previous work13

PEDOT:PTS/Au/TiO2

12.27

0.62

0.51

3.88

previous work15

MoO3/Au/MoO3/PEDOT:PSS/SAMs

5.40

0.62

0.52

1.78

previous work24

MoO3/Au/MoO3/Al2O3/PEDOT:PSS

9.18

0.63

0.48

2.77

previous work28

MoO3/Ag/MoO3

2.72

0.57

0.62

0.96

Other group20

[13] M. A. Rahman et al. Sol. Energy Mater. Sol. Cells, 95(12), 3573-3578.

[15] K. Yang et al. Curr. Appl. Phys., 15, S2-S7.

[20] C. Tao et al. Appl. Phys. Lett., 95(5), 206.

[24] M. Maniruzzaman et al. (2014). J. Nanosci. Nanotechnol., 14(10), 7779-7783.

[28] M. Maniruzzaman et al. (2014). Renew. Energ., 71, 193-199.

Reviewer 4 Report

My comments as stated below:

1- 59% similarity were detected. Please reduce the similarity percentage.

2- please highlight the difference between this work and the others. Especially compared to your previous publications. 

3- A lot of acronymn were used. Please make sure it is mentioned in full at least once when it is first mentioned.

4- On page 4, the Benzoid and Quinoid structure of PEDOT were drawn directly below equation. It should be a figure.

5- The first sentence after figure 6, seems to be missplaced. " (3000 rpm), Glass/MoO3.......structures."

Author Response

We would like to thank the reviewers for their time and valuable comments on our manuscript entitled "Conductive MoO3–PEDOT:PSS composite layer in MoO3/Au/MoO3–PEDOT:PSS multilayer electrode in ITO-free organic solar cells” (Manuscript ID: processes-2188974). We appreciated it as all these comments were precious and helpful for improving the quality of this manuscript. We have carefully studied their feedback and have revised our manuscript to address all the comments and suggestions. We hope this revision will meet your approval to attain publication in the journal of processes.

Our detailed response to the reviewers’ comments is as follows (modification in the revised manuscript is indicated by green highlight):

REVIEWER 4

Comments and Suggestions for Authors:

Authors prepared MoO3–PEDOT:PSS conductive composite (M−PSS conductive), which was used to fabricate ITO-free organic solar cells. This manuscript is well prepared, but the novelty is not high enough. Therefore, I suggest accepting this paper after a major revision. Before resubmitting, the authors should solve the following problems:

Response:

Thank you very much for your review.

Comment 1:

59% similarity were detected. Please reduce the similarity percentage.

Response:

Thank you very much for your suggestion. We agree with your comment and we modified our manuscript for reducing similarity. Currently, the similarity is 37% among them 9% is from our previous work. We have cited all the references and due to common phrases, the similarity is a little high.

Comment 2:

please highlight the difference between this work and the others. Especially compared to your previous publications. 

Response:

Thank you very much for your comment. The difference between this work, our previous work, and other groups are the replacement of ITO electrode by utilizing different electrodes including interlayer. The differences are summarized in the table as follows:

Electrode

Jsc
mA/cm2

VOC
V

FF
%

η
%

Ref.

ITO/PEDOT: PSS

12.47

0.60

0.52

3.91

This work

MoO3/Au/(MoO3-PSS conductive composite)/ PEDOT:PSS

11.99

0.64

0.52

3.97

This work

ITO

7.80

0.67

0.48

2.50

previous work13

PEDOT:PTS

4.90

0.68

0.42

1.40

previous work13

PEDOT:PTS/Au/TiO2

12.27

0.62

0.51

3.88

previous work15

MoO3/Au/MoO3/PEDOT:PSS/SAMs

5.40

0.62

0.52

1.78

previous work24

MoO3/Au/MoO3/Al2O3/PEDOT:PSS

9.18

0.63

0.48

2.77

previous work28

MoO3/Ag/MoO3

2.72

0.57

0.62

0.96

Other group20

[13] M. A. Rahman et al. Sol. Energy Mater. Sol. Cells, 95(12), 3573-3578.

[15] K. Yang et al. Curr. Appl. Phys., 15, S2-S7.

[20] C. Tao et al. Appl. Phys. Lett., 95(5), 206.

[24] M. Maniruzzaman et al. (2014). J. Nanosci. Nanotechnol., 14(10), 7779-7783.

[28] M. Maniruzzaman et al. (2014). Renew. Energ., 71, 193-199.

Comment 3:

A lot of acronym were used. Please make sure it is mentioned in full at least once when it is first mentioned.

Response:

Thank you very much for your suggestion. We agree with your comment and we revised our manuscript according to your suggestion. We use the full name at first and then the acronym.

Comment 4:

On page 4, the Benzoid and Quinoid structure of PEDOT were drawn directly below equation. It should be a figure.

Response:

Thank you very much for your suggestion. We agree with your suggestion. The Benzoid and Quinoid structure of PEDOT is in Figure 2. in the revised manuscript.

Comment 5:

The first sentence after figure 6, seems to be missplaced. " (3000 rpm), Glass/MoO3.......structures."

Response:

We apologize for our mistake. Thank you very much for your correction. Yes, the first sentence after figure 6, was misplaced. " (3000 rpm), Glass/MoO3.......structures." We removed this misplaced part.

Round 2

Reviewer 2 Report

The author answered all the questions。

Author Response

We did not find any queries from reviewer

Reviewer 3 Report

Please perform experiments to investigate the morphology of the prepared electrode. Could the nanomorphology of the film affect the morphology of the active layer deposited atop?

Author Response

Thank you very much for your valuable comments. We agree with your suggestion that the nanomorphology of the film can affect the morphology of the active layer deposited atop. Many researchers already did the same experiment to observe the different quality of active layer on different surfaces [1,2]. The different morphology of active layer on modified electrode or hole transport layer can increase the surface roughness or increased the formation of more extensively ordered structures in thin films of the active layer, resulting in better absorption and charge extraction for improved surface morphology of the active layer film which provided higher power conversion efficiency in solar cells [3, 4].

But due to the lack of time and instrument availability it is quite impossible to perform new experiments to investigate the morphology of the prepared electrode. We extremely apologize for our limitations.

Reference:

  1. Lim, E. (2013). Enhanced photovoltaic performance of P3HT: PCBM cells by modification of PEDOT: PSS layer. Molecular Crystals and Liquid Crystals585(1), 53-59.
  2. Reinspach, J. A., Diao, Y., Giri, G., Sachse, T., England, K., Zhou, Y., ... & Bao, Z. (2016). Tuning the morphology of solution-sheared P3HT: PCBM films. ACS applied materials & interfaces8(3), 1742-1751.
  3. Xiao, T., Cui, W., Anderegg, J., Shinar, J., & Shinar, R. (2011). Simple routes for improving polythiophene: fullerene-based organic solar cells. Organic Electronics12(2), 257-262.
  4. Lim, E., Lee, S., & Lee, K. K. (2011). Improved photovoltaic performance of P3HT: PCBM cells by addition of a low band-gap oligomer. Chemical Communications47(3), 914-916.
